# A New Socially Assistive Robot with Integrated Serious Games for Therapies with Children with Autism Spectrum Disorder and Down Syndrome: A Pilot Study

**DOI:** 10.3390/s21248414

**Published:** 2021-12-16

**Authors:** João Antonio Campos Panceri, Éberte Freitas, Josiany Carlos de Souza, Sheila da Luz Schreider, Eliete Caldeira, Teodiano Freire Bastos

**Affiliations:** 1Postgraduate Program in Electrical Engineering, Federal University of Espirito Santo, Av. Fernando Ferrari, 514, Goiabeiras, Vitoria 29075-910, Brazil; ebertefreitas@ufrn.edu.br; 2Automation and Control Engineering Department, Federal Institute of Espirito Santo, Av. Filogônio Peixoto, 2220, Aviso, Linhares 29901-291, Brazil; 3Postgraduate Program in Biotechnology, Health Sciences Center, Federal University of Espirito Santo, Av. Marechal Campos, 1468, Maruipe, Vitoria 29043-900, Brazil; josianysouza96@gmail.com (J.C.d.S.); sheiladaluz@gmail.com (S.d.L.S.); 4Electrical Engineering Department, Federal University of Espirito Santo, Av. Fernando Ferrari, 514, Goiabeiras, Vitoria 29075-910, Brazil; eliete.caldeira@ufes.br

**Keywords:** socially assistive robot, autistic spectrum disorder, Down syndrome, Serious Games

## Abstract

This work introduces a new socially assistive robot termed MARIA T21 (meaning “Mobile Autonomous Robot for Interaction with Autistics”, with the addition of the acronym T21, meaning “Trisomy 21”, which is used to designate individuals with Down syndrome). This new robot is used in psychomotor therapies for children with Down syndrome (contributing to improve their proprioception, postural balance, and gait) as well as in psychosocial and cognitive therapies for children with autism spectrum disorder. The robot uses, as a novelty, an embedded mini-video projector able to project Serious Games on the floor or tables to make already-established therapies funnier to these children, thus creating a motivating and facilitating effect for both children and therapists. The Serious Games were developed in Python through the library Pygame, considering theoretical bases of behavioral psychology for these children, which are integrated into the robot through the robot operating system (ROS). Encouraging results from the child–robot interaction are shown, according to outcomes obtained from the application of the Goal Attainment Scale. Regarding the Serious Games, they were considered suitable based on both the “Guidelines for Game Design of Serious Games for Children” and the “Evaluation of the Psychological Bases” used during the games’ development. Thus, this pilot study seeks to demonstrate that the use of a robot as a therapeutic tool together with the concept of Serious Games is an innovative and promising tool to help health professionals in conducting therapies with children with autistic spectrum disorder and Down syndrome. Due to health issues imposed by the COVID-19 pandemic, the sample of children was limited to eight children (one child with typical development, one with Trisomy 21, both female, and six children with ASD, one girl and five boys), from 4 to 9 years of age. For the non-typically developing children, the inclusion criterion was the existence of a conclusive diagnosis and fulfillment of at least 1 year of therapy. The protocol was carried out in an infant psychotherapy room with three video cameras, supervised by a group of researchers and a therapist. The experiments were separated into four steps: The first stage was composed of a robot introduction followed by an approximation between robot and child to establish eye contact and assess proxemics and interaction between child/robot. In the second stage, the robot projected Serious Games on the floor, and emitted verbal commands, seeking to evaluate the child’s susceptibility to perform the proposed tasks. In the third stage, the games were performed for a certain time, with the robot sending messages of positive reinforcement to encourage the child to accomplish the game. Finally, in the fourth stage, the robot finished the games and said goodbye to the child, using messages aiming to build a closer relationship with the child.

## 1. Introduction

The increasing use of robots to improve the life quality of children with physical or mental disabilities shows that robotics has taken an important place in contemporary life. These robots usually have cameras, sensors, and mechanisms that allow them to sense their surroundings in addition to be both mobile and autonomous. In this way, therapists can use these robots as tools to obtain parameters of interest in their therapies with children, such as eye contact, physical contact, divided attention, participation in interactive activities, ability to imitate, verbal communication, emotions, etc. Thus, this kind of robot helps improving hypotonia, joint hypermobility, and even vital signs [1].

Currently, several studies are being conducted in the world with the objective of improving therapies with autism spectrum disorder (ASD) and Down syndrome (DS) through socially assistive robots (SARs) [1,2,3]. These studies emphasize that the robots are a significant tool, as they have incited great interest in children. Thus, robots in association with therapists can more significantly stimulate social, cognitive, and motor skills, potentiating the effects of traditional therapies, both in behavioral and in physical rehabilitation of these children [4].

Examples of those robots are PLEO, Keepon, and Paro (nonanthropomorphic) [5,6,7,8], and KASPAR, ROBOTA, and NAO (anthropomorphic) [9,10,11]. All of them are used in ASD therapies, aiming to assist children in demonstrating and perceiving emotions, as well as interacting with others. These SARs help children to express their emotions and offer them the opportunity of human–robot interaction [6].

On the other hand, it is worth mentioning that traditional therapies for ASD and DS children can require up to 40 h of commitment per week [4]. For this reason, different kinds of therapies, provided by several professionals, are commonly associated with high costs. Thus, demand, social commitment, and development of new therapies to improve behavioral and physical aspects, as well as the current progress of technology, especially in assistive robots, are inherent. This work presents an effort in convergence used to develop a new socially assistive robot (with autonomous operation or manually controlled by therapists) with the aim of improving the quality of life of children with ASD or DS. This work first reviews some concepts related to the therapies for DS and ADS, and then shows details of the integrated mini-video projector capable of projecting Serious Games (SG) on the floor or tables to provide more fun and motivating therapies for these children. Results from two out of a few different SGs developed are presented in detail as well. Finally, the data obtained by applying GAS after the children’s interaction with the robot are presented.

The GAS method converts qualitative into quantitative data, allowing evaluating of the success in achieving goals. Thus, the GAS scale choice is due to the fact that it is a flexible assessment scale, chosen in the scientific environment and recurrently used in international studies.

The main research’s hypothesis is that the social robot MARIA T21, integrated with the concept of Serious Games through its image projection capacity, is an active agent that provides greater engagement by children with already-established therapies. In addition, from the various devices available in the robot, it is possible not only to stimulate greater social interaction between the robot, child, and therapist, but also record and assess the evolution of these children undergoing robot-assisted therapies.

It is worth highlighting that this study was approved by the Ethics Committee of UFES/Brazil (number 1.121.638).

## 2. Autism Spectrum Disorder (ASD) and Down Syndrome (DS)

### 2.1. Down Syndrome (DS)

DS, or Trisomy 21, is one of the most common genetic disorders in the world and also the most common cause of intellectual disability. It is caused by extra genetic material on chromosome 21 and affects about 1 in 700 children worldwide [12]. This syndrome causes delayed neuropsychomotor development and hypotonia, and may be associated with other pathologies such as congenital heart defects, hearing and vision problems, cervical spine deviation, obesity, premature aging, and thyroid disorders [13]. Hypotonia is one of the typical features recognized in children with DS, which is related to a significant decrease in muscle tone, i.e., the state of the muscle’s elastic tension at rest. Due to muscle tone, muscle contraction begins after receiving an impulse from the central nervous system [13]. However, in hypotonia, the ligaments’ structures are affected, so they become looser and the joints allow a greater range of motion, which characterizes joint hypermobility, which is also very common in children with DS [14]. Both hypotonia and joint hypermobility affect postural control and the ability to perceive the spatial position of the human body and its orientation. This concept is referred to as proprioception [14]. Therefore, children with DS have inadequate sensory and motor stimuli during their neuropsychomotor development, which slows down their overall developing process, leading to impairments in fine and gross motor skills. For example, it takes longer to control the cervical spine, resulting in a longer time for gait and body balance development [15]. Therefore, it is of utmost importance that therapeutic interventions are carried out at an early stage with both physiotherapists and psychopedadogists. These interventions have shown good results in improving psychomotor development and social skills in children with DS, correcting atypical movement and postural patterns due to inadequate central nervous system function [15].

### 2.2. Autism Spectrum Disorder (ASD)

There are some main features in children’s behavior that may indicate ASD [16]: significant language delay; difficulty in communicating (i.e., making oneself understood and conveying a verbal message); difficulty in interacting with others (leading to self-isolation, inability to play with others, and feeling excluded from society); repetitive, rhythmic, and compulsive or ritualistic behavior. ASD affects approximately 1 in 60 children in the United States [1], which is about 10 times more common than DS. Its diagnosis is essentially clinical, through behavioral observation involving psychologists, psychiatrists, and neurologists. Therapies traditionally used for children with ASD include applied behavior analysis (ABA), which is an important method for mitigating undesirable behaviors such as stereotypies, self-injury, and aggression, as well as promoting the psychosocial skills needed for the child’s development. Additionally, another technique has emerged (Denver model of early intervention), which was developed in the 1980s. It is a naturalistic intervention method that aims to develop and improve the social and language skills of young children in a completely playful way, using the child’s motivation as the main factor for the activity’s development [17]. The various therapies applied to children with ASD intend to break down their typical skills and behaviors into small steps, and each success is rewarded by a reinforcer. This reinforcer plays a vital role in therapy, as it stimulates the children’s abilities [17].

## 3. Socially Assistive Robot (SAR) and Serious Games (SG)

Feil-Seifer et al. [18] defined the acronym SAR as an intersection between assistive robotics and socially interactive robotics. An SAR aims to help or assist a human user to improve their life’s quality, mental health, and socioemotional wellbeing using cameras, sensors, and mechanisms that provide sensitivity, mobility, and autonomy. In this way, therapists can use these robots as tools to obtain different parameters, such as the level of eye contact, physical touch, shared attention, participation in interactive activities, ability to imitate, verbal communication, emotions, hypotonia, joint hypermobility, and vital signs. In [19], the authors applied a system based on multimedia and increased interaction technologies in rehabilitation treatments for children with ASD.

Depending on the therapy chosen by the therapist, the robot can also act as a “teacher” (authority role), as a “friend”, as a “toy” (role of mediator of the behaviors suggested by the therapist), or as an interface that allows the therapist to express him/herself through the robot to communicate verbally or through simulated emotions, and to perform interaction activities [8]. It is worth mentioning that current therapeutic approaches that use SAR draw on recreational resources, particularly educational games. In recent years, several research approaches have been developed to apply these concepts to electronic games in order to provide therapies that can use these resources in a more motivating way. This attempts to create an immersive environment of interactive resources that can aid in both psychomotor and cognitive rehabilitation therapies. These special games are called Serious Games (SG) [20,21].

In addition to psychomotor skills, SGs can cause the child to improve various cognitive skills as these games stimulate problem solving, decision-making, information processing, creativity, and critical thinking [20].

For example, in [22] a video game was proposed for improving verbal skills, in particular prosody, focusing on the design and evaluation of the educational video game, from a point of view about how appealing it is.

The specific attributes of children with ASD and DS have led to an increased development of research on SG with the aim of helping them in distinct areas, such as education, rehabilitation therapies, training and development of new skills, or to complement traditionally applied therapies. By definition, SGs are software developed based on the principles of interactive game design to deliver educational, training, or therapy content, with positive effects empirically proven on users. SGs differ from traditional games used for other purposes than entertainment in that they have positive effects on the user, reducing the cost and time required by the therapist or educator, and increasing the target audience’s acceptance [23].

SGs are typically designed to create an immersive environment based on interactive resources through which the child can perform specific movements, solve problems, and develop new skills, which can help in both psychomotor and cognitive rehabilitation therapies in children with ASD and DS. In addition to immersion, SGs can help the child to perform movements during the psychomotor rehabilitation therapies that improve specific muscle groups, muscle memory, and joints, as well as improve proprioception and body balance. They also show cognitive gains by promoting problem-solving, decision-making, information processing, creativity, and critical thinking [24,25].

Virtual reality-based therapy is one of the most innovative and promising recent developments in rehabilitation technology [26]. This technology allows users to interact with a computer-generated scenario (a virtual world), making corrections and increasing intensity of training while providing feedback [27]. As an example, the game Beesmart, developed for Kinect, improves the user’s day-to-day motor skills [28].

Therefore, the use of SGs is extremely attractive to children because they provide access to a safe, controlled, and predictable environment and also allow a gradual increase in the difficulty level, which can reduce children’s anxiety [29]. The aim of this paper is to show the development and implementation of both the robot MARIA T21 and two Serious Games (one cognitive and the other physical) for interaction with children with ASD and DS. For the evaluation of both SGs we used the “Guidelines for the Design of Serious Games for Children” and the evaluation based on psychology, and for the full evaluation of the system we used GAS.

## 4. The Robot MARIA T21

MARIA T21 is the result of several years of research and development at UFES/Brazil in the field of robots for interaction with children. Great progress has been made in terms of design, structure, intervention protocols, and support from medical professionals. Thus, the new robot MARIA T21 was designed using proprietary technologies to help children with ASD and DS with both basic training and assessment of their dynamic characteristics. Figure 1 shows the robot interacting with children with ASD and DS.

MARIA T21 is portable, has an adjustable height from 1.10 m to 1.40 m (which allows a better adaptation to the child’s needs) and can be adapted to different therapeutic proposals, acting as an authority figure (such as a teacher) or as a friend or toy. In its development, we sought to ensure that the robot could directly help not only children with ASD and DS, but also family members and therapists. All of MARIA T21’s devices are integrated into its physical structure, which is covered with a touch-sensitive coating that adds the ability to sense and respond positively to touch (which is very important for children with ASD, who often have an aversion to physical contact). In addition, such as the aforementioned, MARIA T21 can project SGs onto the floor or table while interacting with children, guiding and encouraging them through music and prerecorded verbal statements (artificial voice). All the interaction data are recorded after each interaction. The projected SGs allow the child to interact with the robot through cards, textures, and figures. The robot has arms with two degrees of freedom, which is used as a tool to reinforce the interaction with the child and further explore the SGs. After each interaction, the robot issues a report on the child’s performance, which makes it possible to evaluate the child’s development or resistance to the intervention protocol established by the therapist. The robot MARIA T21 can express different emotions on the face, depending on the interaction with the child: great joy, happiness, sadness, or fear. The main components for the SG’s application are listed below.

### 4.1. Software

PyGame: A set of libraries that allows the SG creation graphically. It uses the Python language and allows the creation of various graphical screens, as well as other interface features such as assorted images, animations, and sounds. The advantage is that the code is written entirely in Python and is distributed under the Lesser General Public License (GNU), which allows it to be used in both commercial or free software.

Robot operating system (ROS): It is open-source and allows the development of robotic systems, so that it is possible to manage device drivers and other components used to abstract the hardware. It contains visualization and simulation tools, management in client–server architecture, communication protocol, and simplifies the messages exchange between different processes [30]. In this work, it is used to provide the necessary middleware for the local network development that performs the communication between the PC or the therapist’s tablet and the robot.

### 4.2. Hardware

Mini-video projector: It projects images up to 2 m wide at a distance of 80 cm from the screen. It is portable and completely wireless, with built-in sound, 450 lumens brightness, and 55 W maximum power consumption (LG model PH450U).

Laser sensor: The main application of this sensor (model LiDAR—light detection and ranging) is to determine the child’s pose (position and orientation) in the projections in order to allow real-time interaction between the child and the robot.

Video cameras: One camera (model GoPro Hero 4) is used to capture the child’s facial expressions to allow the child’s emotions analysis and to obtain dynamic parameters during the robot’s interaction. The other camera (model Logitech C920 PRO) is placed on the robot’s head to capture images of the playing cards containing QR code identifiers.

Finally, the robot has a laptop on board (model Dell Gamer G3-3500-A40), which is used to communicate with external devices via a wireless router (model Archer C6 Dualband).

## 5. Methodology

Given the sanitary issues imposed by the COVID-19 pandemic in 2020, the sample of children was limited and included eight children aged 4 to 9 years old: one child with typical development, one with Trisomy 21, both female, and six children with ASD, one girl and five boys. The established age range of 4 to 9 years excluded children under the age of 4 years so that there was minimal cognition necessary to play the Serious Games with little or no assistance. The upper 9-year interval is limited by availability questions, reinforced by the impediments generated by the pandemic, especially with typically developing children with Down syndrome. The tests were executed out partly in a countryside region and partly in a metropolitan area, in order to expand socioeconomic diversity. The inclusion and exclusion criteria were previously defined and sent to the health professionals of the partner institutions, who selected the children in their set who met the requirements. There was no participation of researchers.

For the experiments of this study, the games were created with all their possible events, characters, awards, and stories. During the games idealization and development, theoretical foundations and consultations with health professionals were always sought in order to better adapt the SGs to real-life use conditions and their purpose, with the aim of maximizing the content explored.

During the interviews accomplished with some health professionals, it was observed that some simple practices applied in conventional therapies could be adapted in games’ conceptualization, for example, a therapy applied for the development of children’s gait consisting of several footprints drawn by the therapist on the floor, which tries to encourage children to correctly perform walking movements. This concept was very important in some of our games’ development, as the robot can easily project these footprints in various positions, thus expanding the concept explored by therapists in conventional therapies.

The SGs developed for this study were “Walking on the Rope”, “Jump Rope”, “Hopscotch”, “Force Hammer”, “Music Therapy”, “Let us Dance!”, “Magic Carpet”, “What is the Card?” and “Animal Detective”. All their graphical interfaces are shown in Figure 2, Figure 3, Figure 4, Figure 5, Figure 6, Figure 7, Figure 8, Figure 9, Figure 10 and Figure 11.

Figure 2 presents the game “Walking on the Rope”. This game aims to train children’s balance, proprioception, and motor coordination. The images are projected on the floor, with a cliff and a rope on which the child must walk to reach the other side of the cliff. This game consists of three different levels: in the first, the child must walk through the rope’s image while holding a soft ball until he/she reaches the other side of the cliff. In the second level, the child must run on the rope holding the ball until they reach the other side of the cliff. Pictures of birds appear along the way to make the task more difficult. In the third level, the child must run over the rope’s picture with the ball until he/she reaches the other side of the cliff. While running, a limited part of the rope’s image starts flashing, disappears after 5 s, and then resumes after 3 s.

Figure 3 presents the game “Jump the Rope”, which also aims to train children’s balance, proprioception, and motor coordination, and involves the projection of a moving rope onto the ground, over which the child is asked to jump. There are four stages: in the first level, the rope’s image is projected onto the floor for the child to jump with free time. In the second level, the time for the rope’s image to appear on the floor for the child to jump is shortened. In the third level, the rope’s projection speed is the same as in the first level, but with an additional projection of a ball thrown to the child so that he/she can catch it while jumping over the rope. In the fourth stage, a virtual child is added to the projection to perform the activity at the same time as the child to encourage competition.

Figure 4 presents the game “Hopscotch”, which aims to train balance, proprioception, and motor coordination by projecting the image of hopscotch onto the floor. It consists of four stages: In the first level, the child must jump over the hopscotch game, according to the number (projected by the robot) on the floor. In the second level, the child must jump over the hopscotch (in this case, instead of numbers, footprints appear, some facing forward, some facing right, and some facing left). In each hopscotch square, the child must jump with his/her feet in the direction indicated by the footprints. On the third level, the time for the level 2 task is shortened. On the fourth level, a virtual child is added to the projection to encourage competition.

Figure 5 presents the game “Force Hammer”, which aims to train children’s postural balance, proprioception, and motor coordination, in addition to their modular stereotypes and divided and joint attention training. It consists of projecting images of a target so that the child engages with both feet simultaneously after a jump, and a column that indicates the score, together with a dot marker that moves along this column. It consists of four stages: In the first level, the child must jump on the marker with both feet and observe the score over it. In the second level, the child has to perform the jump while the robot asks him/her, by voice command, to perform another task together, such as catching a ball (thrown by the therapist). In the third level, the child must perform the jump with unimodal support. In the fourth level, a virtual child is added to the projection to encourage competition.

Figure 6 presents the game “Music Therapy”, which is also designed to train proprioception, motor coordination, balance, and divided and joint attention. In this game, images related to the narrative are projected, such as forests, beaches, tasks, and others. Then, the robot draws the scenarios included in the narrative on the floor by projection and moves through the environment near the child. The child is asked to walk alongside the robot and explore the projected scenarios, completing the desired tasks. Examples of such tasks are as follows: (1) the robot plays an exciting song while the child explores the projected environment (forest) and encounters one or more animals (that make sounds) at a predetermined time. The robot can also give the child a different type of task (e.g., instructions to escape if it is a large and potentially dangerous animal, such as a jaguar; feed the animal; or take the animal to a specific location). (2) Images of landscapes, calm and happy moments, while the robot is playing classical instrumental music (and observing the child’s behavior, including motor assessment). (3) Projection of geometric shapes with different colors on a table, in rhythm with the song. In addition, the therapist shows a sequence of touches on the geometric shapes and asks the child to repeat the sequence.

Figure 7 presents the game “Let us Dance!”, which is also designed to train proprioception, balance, motor coordination, and divided attention. In this game, an avatar is projected to perform choreography while playing a nursery rhyme that the child must follow. To support this task, footprints are projected on the floor alongside the avatar, on which the child must take steps to direct his/her movements to more closely resemble the avatar’s choreography. The song choice is based on the dance style and the melodies that require specific moves, such as raising hands, jumping with one foot, touching the head, turning the body, and others.

Figure 8 presents the game “Magic Carpet”, which aims to train proprioception and motor coordination, stimulate contact with new textures, and encourage touch by children. A mat is used on the floor, which has five different areas with different textures, all equipped with load cells for foot pressure analysis. Each area acts as directional controls, depending on the projection in front of the child and the robot’s voice commands, which lead the child on an adventure. The child must avoid obstacles such as birds, airplanes, and clouds (if the adventure takes place in the sky), as well as rocks, fish, and water streams (if the mat travels through an aquatic environment). For example, if the child is standing on the mat’s left side, he/she can move the virtual mat to the left. The center of the mat is a neutral area, corresponding to the command to keep the virtual mat’s course stable. Depending on the obstacle in the path, the child can alter the course of the mat by stepping on one of the four available areas to move away from the obstacle. If the child does not move, the mat collides with the obstacles. The robot stores information about the stepping time in each mat area at all times and also records the child’s facial expressions at every moment.

Two of these SGs are detailed in this work: “What is the Card?” (Figure 9 and Figure 10) and “Animal Detective” (Figure 11), as shown below.

### 5.1. What Is the Card?

This game (Figure 8) aims to develop the child’s knowledge about the first five numbers and vowels, as well as the motor coordination, and trains joint attention. At the beginning, the robot MARIA T21 introduces itself and invites the child to play. The game has three stages, which are described below.

**Level** **1:**This level contains only the first five numbers, with colors for each number: 1 in blue, 2 in red, 3 in green, 4 in yellow, and 5 in black.**Level** **2:**This level follows the same principle as the previous one, but now the child has to identify the cards with their respective vowels and show them to the robot. These are shown as A in blue, E in red, I in green, O in yellow, and U in black.**Level** **3:**In this third and final level, the robot projects numbers and vowels randomly. Thus, the child must not only be able to identify the individual numbers and show them to the robot, but also recognize the colors between numbers and vowels. The action sequences must be selected according to the therapist’s commands.

The therapist interface has the following commands: Robot self-introduction, Prompt to play, Level 1, Level 2, Level 3, and Exit. Figure 9 shows the game interface projected on the floor while the robot MARIA T21 guides, motivates, and presents characters to the child. Hit counters are displayed at the game’s left top and right bottom, with a new mark added for each correct number or vowel. The levels are advanced only after ten attempts with correct answers, together with verbal and visual congratulations at the end of the level. Figure 10 shows the ten cards available for the children to play with.

In this game, the numbers order is random on each run. At the end, the robot generates a report with the interaction parameters to facilitate the therapists’ analysis, such as the number of correct answers in each level, the total amount of game projections, the total number of correct answers, the total number of errors, the execution time from start to finish, the game interruptions (if any), and the child’s reaction time to show each card.

### 5.2. Animal Detective

The game’s goal is to keep a detailed record of the child’s gait by recording his/her movements during the game’s exploratory stretches. The game not only allows the subsequent exhaustive evaluation of the child’s gait, but is also playful, interactive, works with divided attention, and presents animals and their respective sounds. At the beginning of the game, the robot tells stories to contextualize the child’s need to walk in a particular projection for each difficulty game level. For example, in the first level, the robot needs help finding five animals hidden in the bushes during a visit to the zoo. While performing this game, the robot’s facial expression changes from sad (at the beginning of the game, in the moments when the animals are hidden), to happy (when some animals have already been found), and to very happy (when he/she finds all the animals of the given level, making their respective sounds while the robot says their names, presenting them). In each level, the animals’ sizes are reduced, forcing the child to go through the entire game projection more and more carefully. In addition, the environments in which the animals hide are increasingly full of information and details, from a green area with homogeneous bushes in a zoo (on Level 1) to a school with trees, benches, and toys scattered on the floor (on Level 5). In this case, the animals are different to match the new environment. On the other hand, the animals’ positions are random, and not even the researchers can predict them in the projection. Figure 11 shows the interface for Level 1 of this game, with four of the animals already discovered. At every moment during the game, the robot stores data about the child’s pose (via laser sensor), randomizes the animals’ positions, and plays some of their characteristic sounds to help the child identify them.

### 5.3. SG Pilot Test with Child with DS and Typically Developing Child

First, a pilot experiment on the interaction between children and robots was conducted with two girls, one with DS (12 years old) and the other with typical development (5 years old). Protocol consisted of inviting the child with her mother to the classroom, where the physiotherapist and the biologist explained the experiment to the child, and invited MARIA T21 into the room. The robot then executed self-presentation under command of the researcher and invited the child to play. However, both girls initially showed fear of the robot, which could be related to the dynamics in which the robot was suddenly introduced them, in a surprise mode (the children were already in the experimental room when the robot suddenly appeared). The child with DS interacted with the robot for about an hour. During that time she played the game “What is the Card?”, showing curiosity and interest at the game (Figure 12), but found it hard to understand the robot’s commands to play the game. She also showed an affectionate relationship with the robot, calling it “friend”. At the end of the session she cried because she did not want to leave.

In the second game (“Animal Detective”), the child felt comfortable to ask the robot about personal information, inviting it for a walk, and fulfilling the goal suggested by the game (Figure 13). However, the girl resisted walking over the projection, which hindered this game’s interaction development.

The typically developed child interacted with the robot for about twenty minutes, just playing the game “What is the Card?”. She showed curiosity in doing so, and told her mother that she would like to play with the robot again.

### 5.4. Experiments with Children with ASD

The experiments were carried out with the participation of six children with ASD for two days, assisting half of the participants each day. Among the participants, there were one girl and five boys who met the following criteria:Inclusion criteria:Age from 4 to 12 years.Perform therapy for at least 1 year.Have a diagnosis of ASD.Exclusion criteria:Not having osteomyoarticular deficiency.Not having too many stereotyped movements.

The protocol was carried out in a child psychotherapy room in the presence of a research group composed of two electrical engineers in charge of controlling the robot, a PhD student in biomedical engineering, a physiotherapist, and a biologist. During the experiments, the therapist stood next to the child and the robot in the interaction room. In contrast, the other researchers and the person in charge of the child sat away from the interaction environment. In this environment, three video cameras were placed in strategic locations for the researchers to consult later, one on each side of the interaction room and another behind the robot to capture the child’s front. It is worth commenting that the robot has two video cameras, one placed at the head and the other at the bottom to capture the child’s legs movements. The protocol begins with the child entering the room, then, the robot enters the room, introduces itself, and asks about the child’s name, age, and if he/she would like to play. The first few minutes were used to record the child’s approach to the robot and vice versa, and to assess the children’s responses, such as proxemics, attention to phrases emitted by the robot, and touching the robot. In the protocol’s second phase, the child was invited by the robot to play the game “What is the Card?”. Up to 10 min was reserved for the game, which could be interrupted before the expiration time if the child was not interested. The therapist and caregiver were free to explain the games to the child before they began. In the sequence, the robot sent positive reinforcement messages, wishing the child success, and encouraging him/her when he/she made mistakes. In the third moment, the child was asked to play the game “Animal Detective” for 15 min. This longer time was set under the consideration of the child’s possible resistance to run on the game’s projection in the correct orientation with respect to the robot’s camera. At the end of the second game, the robot asked the child if he/she had enjoyed and which game he/she liked most. The interaction ended with the robot saying goodbye to the child and thanking him/her for playing together. Then the therapist escorted the child and caregiver to the exit. The six participants in this study are referred to by their initials to preserve their identities: L.M.B., age 4, mild autism; L.A.S., age 7, mild autism; G.S.V., age 8, mild autism; P.C.G., age 9, mild autism; H.V.S., age 9, moderate autism; R.R.C.F., age 8, mild autism.

## 6. Results

In this study, five objectives were defined and given equal weighting (equal to 1). To calculate the success in achieving the proposed objectives, we used the method GAS [31] applying the equation
(1)$$T=50+C_{x}\sum x_{i}$$
where $C_{x}$ is the number of general objectives’ coefficient, which, in this case, for five general objectives, corresponds to 3.01; and $x_{i}$ corresponds to the GAS score obtained for each objective. *T* equal to 50 corresponds to the expected performance level; *T* greater than 50 reflects performance above the expected level; and *T* less than 50 reflects performance below expectations [32]. Table 1 shows the evaluated objectives and the respective scores of the method GAS. On the other hand, Table 2 and Table 3 show the interaction’s results, where it is shown that the six children with ASD performed significantly above expectations.

## 7. Discussion

Although this study has a preliminary character, and the sample of children was reduced due to the pandemic, the experiments showed that the robot MARIA T21 was widely accepted by the research’s participating children, such as shown through the GAS scale. Thus, it was possible to observe positive responses concerning interactions between the child and the robot, such as eye contact, physical, and verbal contact. In addition, the robot proved to be a facilitator for the therapist’s interaction with the child, mainly through the tasks verbally proposed by the robot for the execution of the Serious Games. It is worth highlighting the children’s emotional interaction with the robot at the end of the therapies, and it is possible to observe in the children typical reactions of a close friendship, such as hugs, fondness, and phrases of affection. Thus, when compared to other research about the interaction of children with ASD or DS with social robots [1,2,3], the robot MARIA T21 provides significant gain in the field of assistive robotics applied to ASD and DS, by virtue of its projection system integrated into the concept of Serious Games. In addition, this integration optimizes the entire process, as it is a single piece of equipment. Moreover, the child’s immersion and engagement in therapy becomes even greater, as the robot MARIA T21 participates actively in the triad of therapist–robot–child interaction. Once the ability of the robot MARIA T21 is observed to produce greater engagement of children in therapies, facilitating the interaction between the therapist and the child, it is expected that the incorporation of social robots with such characteristics in intervention protocols and guidelines for children with ASD and DS will generate a therapeutic gain.

According to Almeida et al. [23], there are some studies that have dealt with the development of SGs for children with behavioral and psychological problems, especially when it comes to the applicability of these theories. They also showed how the psychological principles were used in the development of four SGs for children with ASD. In our SGs, we used positive reinforcement through stimulating sentences emitted by the robot with an artificial voice, in addition to changes in its face that tell the child that the answer is correct (in the game “What is the Card?”). We did not use any form of punishment in the game to avoid aversive feelings such as frustration and anxiety. In addition, we sought to develop SGs in such a way that adapting the games to different stereotypes or motor and cognitive limitations was possible and easy. In this way, our games can be changed by editing the display time of numbers and cards, allowing the games’ sounds to be turned on and off at any time during their execution, presenting them quickly to avoid scattering, which stems from the child’s fatigue and is exacerbated by hyperactivity or concentration trouble, and using specific themes. In evaluating the quality of the two games developed here, they share thirteen good features (which characterize a good game) with the twenty-four features listed by [23]. However, the game “What is the Card?” has fourteen of the twenty-four features, and the game “Animal Detective” has eighteen of these features. Valenza et al. [12] proposed a different form of evaluation in the use of SGs through guidelines for the design of SGs for children, which were divided into four elements (interface/input and output element, content, and control) and include a total of 40 guidelines. The guidelines identify SGs’ elements that need to be improved to make it child-friendly. For the game “What is the Card?”, a total of 25 of the 40 guidelines were met, including four for the input elements (4 out of 6), eleven for the output elements (11 out of 19), eight for the content elements (8 out of 12), and two for the control elements (2 out of 3) [12]. It is worth mentioning the unmet requirements of the output element, which state that the user’s interface must have a realistic look when targeting children aged 7–9. This is important information, although the development of typically developing children is not the same as children with ASD or DS. For the game “Animal Detective”, a total of 30 guidelines were consistent with the study of [12], namely, five of the input elements (5 out of 6), thirteen of the output elements (13 out of 19), ten of the content elements (10 out of 12), and two control elements (2 out of 3). Here, the output element’s unmet guidelines are highlighted (which correspond to the game goals in question), which dictate that “attention and concentration efforts are minimized” and “recognition is preferred over recall”. It should be noted that the proposed guidelines for the development of SGs for children, while of great value, are not fully compatible with the specificities of children with ASD and DS. This is especially true for the guideline “explore cooperative use” of the input elements, which the authors believe can increase productivity and satisfaction through collaboration during games, and is a common problem found in dealing with children with ASD.

As for the limitations of the study, it is noticeable that the main difficulty encountered in this research was during the selection of the sample of children as a result of the existing health issues imposed by the COVID-19 pandemic during this study’s period of protocol tests execution, considerably reducing the options of available partner clinics. However, due to the current relaxation of social distancing rules, new partnerships are being signed. Currently, a new testing protocol is being carried out with 15 new children, all with ASD, in order to generate new data for future publications.

## 8. Conclusions

Developing SGs for a socially assistive robot requires a lot of effort and resources, and also requires constant testing and adjustments, either in the code or in the visual part (interface). In addition, one needs to decide on the tools and languages in advance so that it is not necessary to redevelop the game. Despite their limitations, Python and Pygame have proven to provide excellent basic support for the game development, in addition to their compatibility with ROS. Although the games “What is the Card?” and “Animal Detective” have simple functionalities that meet the expectations of building game-based therapies, they seem to have the potential to meet the basic premise of being an important tool for recreational therapy collaboration for children with ASD and DS, allowing them to experience more mind–body interaction and greater participation in therapies. As the robot MARIA T21 provides more opportunities for interaction, new games can be developed to explore its ability to move arms and head articulation, its sensitivity to physical contact, the capacity to display various facial expressions and emit sounds (music or voice), and its ability to demonstrate feelings through the robot’s face. All these capabilities are already integrated into the structure of the robot and will soon be explored through new experiments with children with ASD and DS. In conclusion, it is highlighted that this new robot was built with the lab’s proprietary technology and can be an important tool for recreational therapy through the SGs developed for the robot, compared to static games, as they provide greater mind–body interaction and promote greater therapy adherence in children with ASD and DS. In addition, this research is expected to have social, therapeutic, and scientific relevance and also to improve and optimize the provision of care services for these children.

## 9. Patents

There is a patent requirement in the patent’s sector of the Federal University of the Espírito Santo, Brazil.

## Figures and Tables

**Figure 1 sensors-21-08414-f001:**
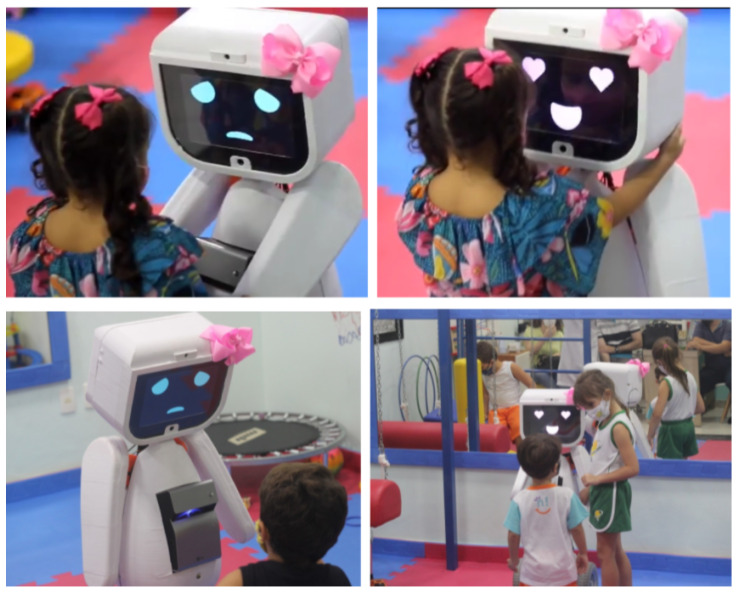
MARIA T21 interacting with children with ASD and DS, and demonstrating its ability to express emotions in the face.

**Figure 2 sensors-21-08414-f002:**
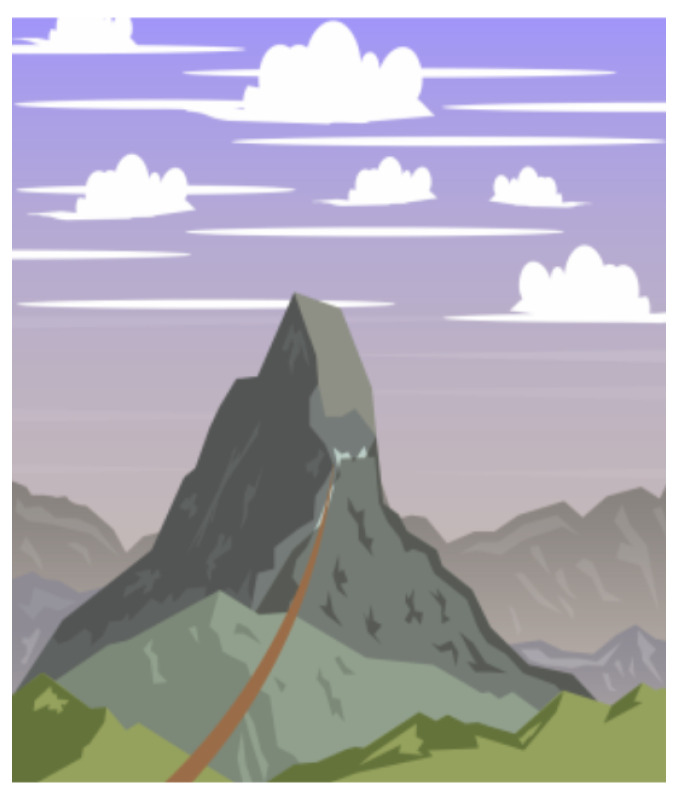
Interface of the game “Walking on the Rope”.

**Figure 3 sensors-21-08414-f003:**
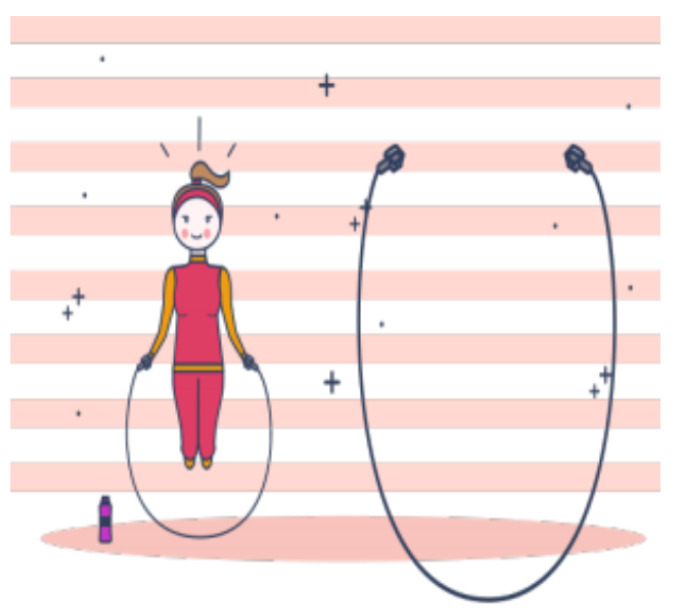
Interface of the game “Jump the Rope”.

**Figure 4 sensors-21-08414-f004:**
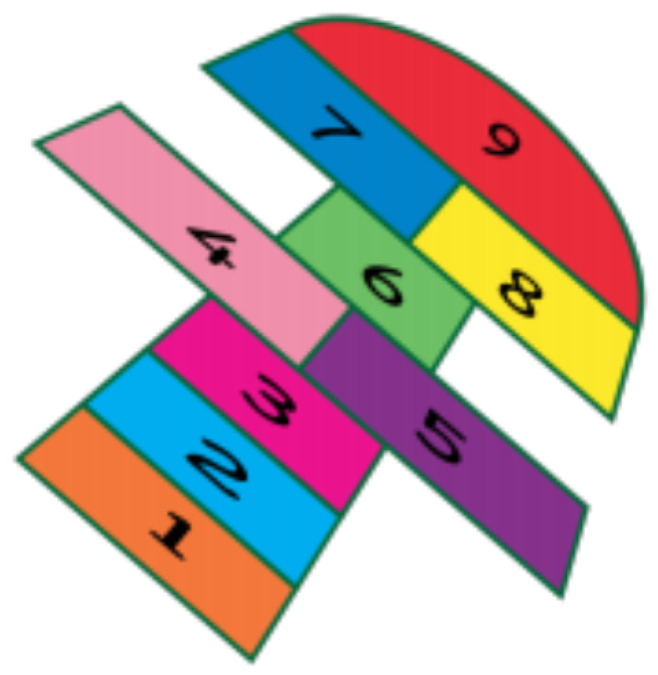
Interface of the game “Hopscotch”.

**Figure 5 sensors-21-08414-f005:**
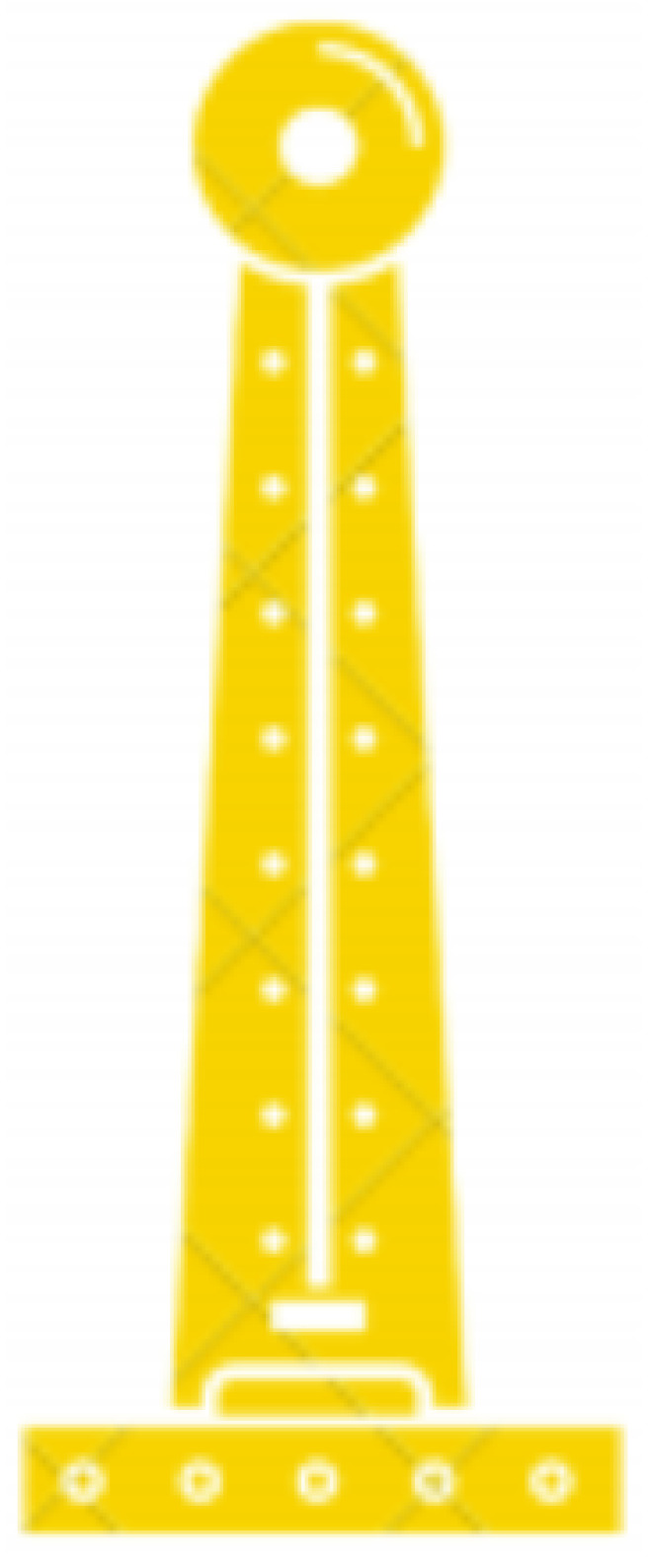
Interface of the game “Force Hammer”.

**Figure 6 sensors-21-08414-f006:**
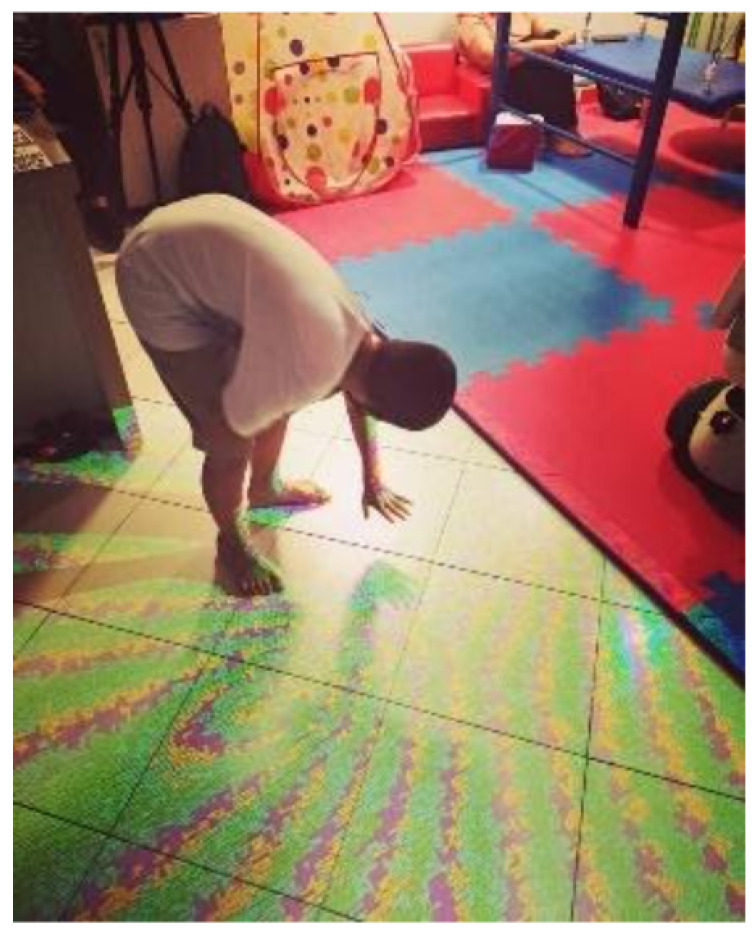
Interface of the game “Music Therapy”.

**Figure 7 sensors-21-08414-f007:**
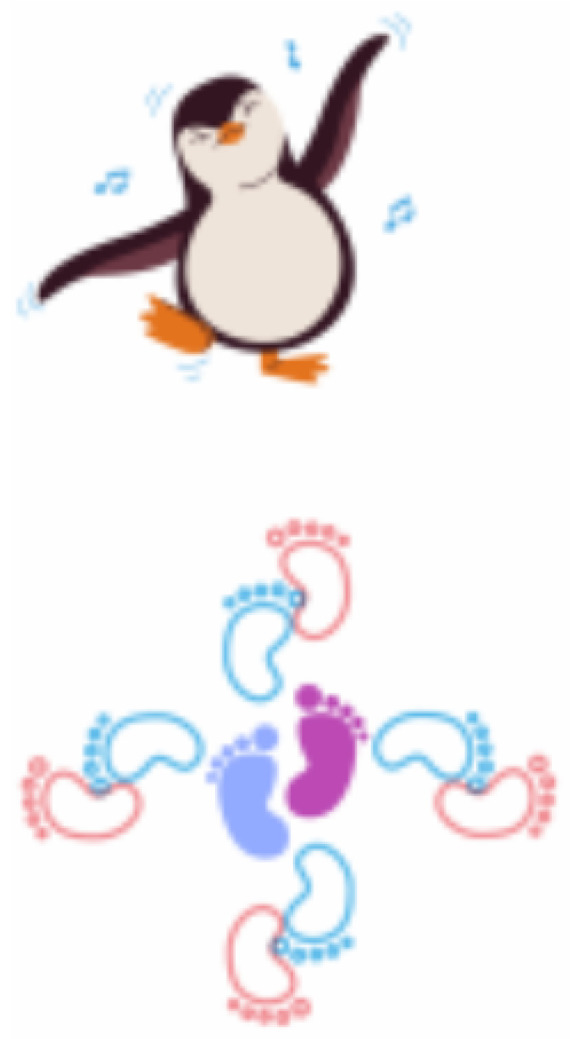
Interface of the game “Let us Dance!”.

**Figure 8 sensors-21-08414-f008:**
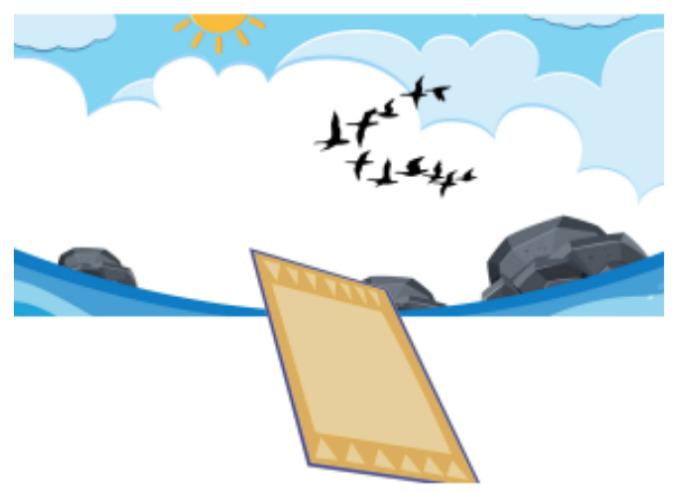
Interface of the game “Magic Carpet”.

**Figure 9 sensors-21-08414-f009:**
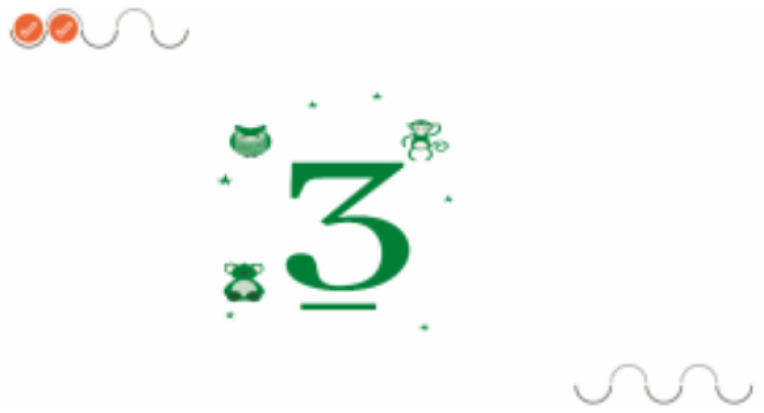
Interface of the game “What is the Card?”.

**Figure 10 sensors-21-08414-f010:**
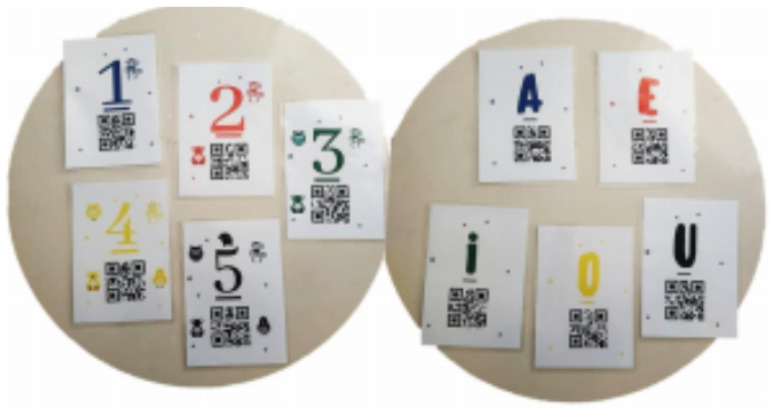
Cards approximately 14 cm long containing QR codes that children must show to the robot. The camera on the robot’s head detects if the card is correct based on the QR code.

**Figure 11 sensors-21-08414-f011:**
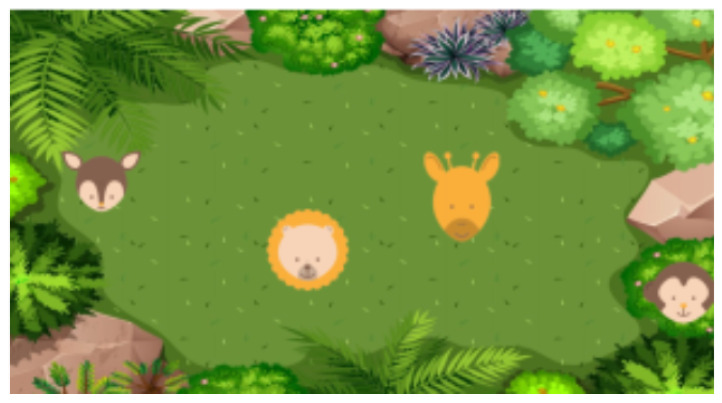
Interface of the game “Animal Detective”, Level 1.

**Figure 12 sensors-21-08414-f012:**
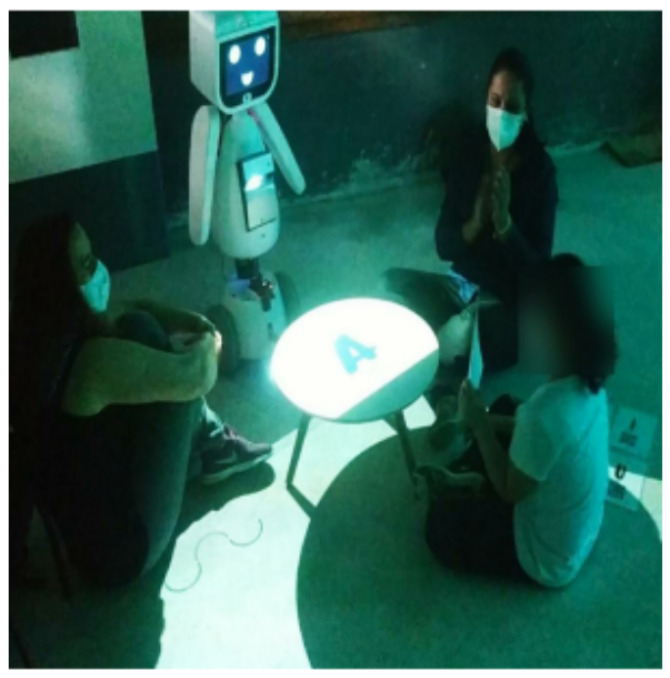
Girl interacting with the robot during the game “What is the Card?”.

**Figure 13 sensors-21-08414-f013:**
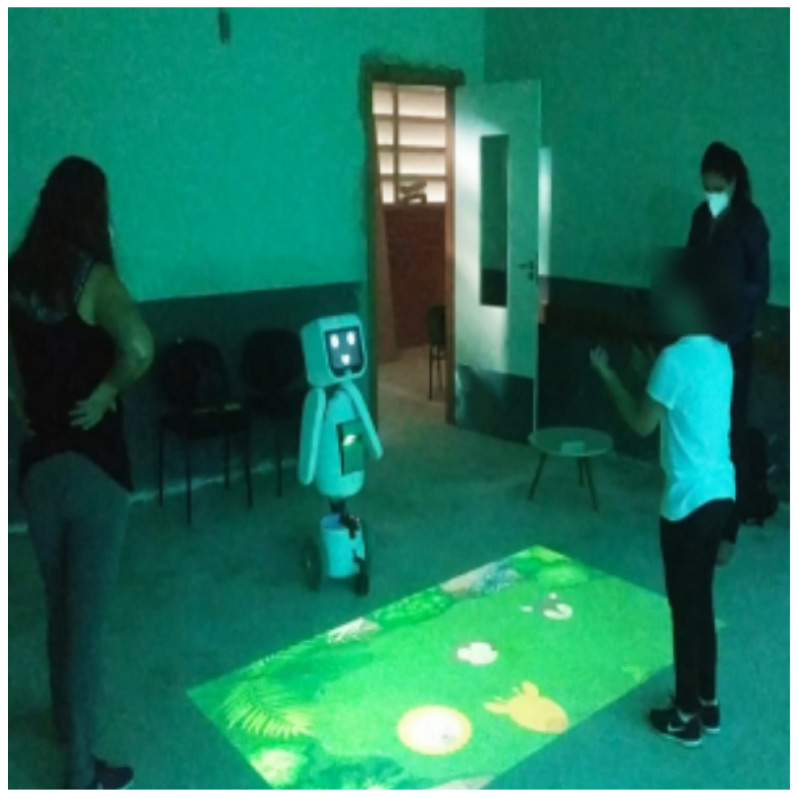
Girl interacting with the robot during the game “Animal Detective”.

**Table 1 sensors-21-08414-t001:** GAS method for the five objectives of the experiment.

Performance and Score	Look at the Robot	Touch the Robot	Talk to the Robot	Play the Games	Interact with the Mediator
Much worse than expected result (−2)	Look at the robot for less than 30 s and show repulsion	Do not touch the robot	Do not perform any dialog with the robot	Do not play the games	Seem to do not understand the mediator’s commands and do not carry them out
Worse than expected result (−1)	Look at the robot for less than 30 s and not show interest	Touch the robot for less than 5 s	Keep the dialog for at least 10 s	Play the games for a few seconds	Seem to understand the mediator’s commands, but do not carry them out, even when encouraged
Expected outcome (0)	Look at the robot for more than 30 s and keep visual contact with the games projections	Touch the robot for more than 5 s	Keep the dialog for more than 10 s	Finish at least one of the games	Understand the commands and carry them out, encouraged by the mediator
Better than expected result (+1)	Look at the robot for more than 30 s and pay attention to the games projections	Touch the robot for more than 5 s and pay attention to the games projections	Keep the dialog for more than 10 s and ask questions to the robot	Finish at least one of the games and perform most part of the other games	Understand the mediator’s commands and carry them out spontaneously
Much better result than expected (+2)	Look at the robot for more than 30 s and go towards it spontaneously	Touch the robot for more than 5 s and play with it	Keep the dialog for more than 10 s and include the mediator	Finish all the games	Understand the commands and perform them spontaneously and together with the mediator

**Table 2 sensors-21-08414-t002:** Average value of T of the GAS method for evaluating child–robot interaction.

	GAS
Children with ASD	77.09

**Table 3 sensors-21-08414-t003:** Values of the GAS method for each of the children (the average value of all the children is 9).

Child	GAS Value
L.M.B., 4 years	4×(+2)+0=8
L.A.S., 7 years	5×(+2)=10
G.S.V., 8 years	4×(+2)+0=8
P.C.G., 9 years	5×(+2)=10
H.V.S., 9 years	4×(+2)+0=8
R.R.C.F., 8 years	5×(+2)=10

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
