# Peer review of "A New Socially Assistive Robot with Integrated Serious Games for Therapies with Children with Autism Spectrum Disorder and Down Syndrome: A Pilot Study"

_sensors, 2021, doi:10.3390/s21248414_

Round 1

Reviewer 1 Report

General comment:

The article is interesting but I cannot evaluate the IT technical part due to a lack of skills.

I think it might be better to divide it into 2 separate articles and better investigate the use of the new tools proposed, in an article on ASD and in another article on DS.

The designs of the two researches are very weak, the samples not well constructed.

Title:

Add that these/this are/is pilot studies/study

Abstract/summary:

---clearly state the aims

---state the sampling strategy, sample size, response rate and main sample characteristics

---describe the design of the pilot studies/study

Introduction:

---describe any necessary background information about the setting for the study

---justify the choice of measures and the sampling strategy

---clearly state the research questions/hypotheses

Methods:

---clearly stated how the sample was obtained in such a way as to be able to judge its representativeness

Discussion:

---begun the discussion with a summary of the main findings

---relate the findings to previous research in terms of whether they support or fail to support the conclusions of that research

---explain how the findings reflect on theory, practice or policy formulation

---examine the limitations of the study, addressing issues such as sample size, sample representativeness, measurement error, measurement bias, whether any intervention was successfully implemented, whether there was contamination between different intervention conditions and ability to generalize from the findings

Author Response

To: Reviewer (Manuscript ID: sensors-1481249) 

Dec 05, 2021 

Dear Sir or Madam, 

We would like to thank you for the comments in the previous revision that allowed us to write a better text. 

Thank you. 

Sincerely, 

João Panceri and colleagues 

Federal University of Espírito Santo

Reviewer 2 Report

The authors present a timely and interesting research. It is always good to read about newly developed assistive technologies / methods.

The English of the study is fine, only a minor spellcheck is required.

The number of references (20) must be improved. More references about the state of the art of SARs and serious games are needed.

Abbreviations should not be used in the abstract, and they should be defined in the main text.

Introduction should be section 1 (instead of 0). This means that all section number should be incremented as well.

Line 34: "Thhis ork" -> "This work"

In section 3.1. and 3.2. each list element should be started in a new line for better readability.

In the Methodology section, the authors wrote "During the games idealization and development theoretical foundations and consultations with health professionals were always sought in order to better adapt the SGs to real-life use conditions and their purpose with the aim of maximizing the content explored." The reviewer is interested in these consultations. They should be elaborated in a paragraph or two.

Figures 2 - 8: the text of the figures is quite long. The reviewer believes that these games should be detailed in a new subsection instead in the figures' text.

The references are not in the style of the Sensors journal. For example: journal names should be abbreviated and semicolons should separate different authors names. In some lines the journal names are well written, but in some lines they are not. There are other formatting errors as well.

Author Response

(The authors gave the same response as above.)

Round 2

Reviewer 1 Report

Dear authors, congratulations on your work. I see that the changes I recommended have been made. The work done is interesting and innovative. As for the part on children with autism, I would like to point out the following work that may be useful to consult and to cite.

Magrini, M.; Curzio, O.; Carboni, A.; Moroni, D.; Salvetti, O.; Melani, A. Augmented Interaction Systems for Supporting Autistic Children. Evolution of a Multichannel Expressive Tool: The SEMI Project Feasibility Study. Appl. Sci. 2019, 9, 3081. https://doi.org/10.3390/app9153081

Best Regards

Author Response

To: Reviewer (Manuscript ID: sensors-1481249)

Dec 08, 2021

Dear Sir or Madam,

We would like to thank you for the comments and help in the previous revision. It has enriched this work.  We really enjoyed the reference you sent us, and we've added a short citation in the line 143. For our future works, we'll use it as a reference. 

Thank you.

Sincerely,

João Panceri and colleagues

Federal University of Espírito Santo